# Industrialization as a source of heavy metals and antibiotics which can enhance the antibiotic resistance in wastewater, sewage sludge and river water

Jakub Hubeny[1], Monika Harnisz[1]*, Ewa Korzeniewska[1], Martyna Buta[1], Wiktor Zieliński[1], Damian Rolbiecki[1], Joanna Giebułtowicz[2], Grzegorz Nałęcz-Jawecki[3], Grażyna Płaza[4]

1 Department of Water Protection Engineering and Environmental Microbiology, Faculty of Geoengineering, University of Warmia and Mazury in Olsztyn, Olsztyn, Poland, 2 Department of Bioanalysis and Drug Analysis, Faculty of Pharmacy, Medical University of Warsaw, Warszawa, Poland, 3 Department of Environmental Health Sciences, Faculty of Pharmacy, Medical University of Warsaw, Warszawa, Poland, 4 Faculty of Organization and Management, Silesian University of Technology, Zabrze, Poland

* monika.harnisz@uwm.edu.pl, monikah@uwm.edu.pl

**Data Availability Statement:** All relevant data are within the manuscript and its Supporting information files.

## Abstract

The spread of antibiotic resistance is closely related with selective pressure in the environment. Wastewater from industrialized regions is characterized by higher concentrations of these pollutants than sewage from less industrialized areas. The aim of this study was to compare the concentrations of contaminants such as antibiotics and heavy metals (HMs), and to evaluate their impact on the spread of genes encoding resistance to antimicrobial drugs in samples of wastewater, sewage sludge and river water in two regions with different levels of industrialization. The factors exerting selective pressure, which significantly contributed to the occurrence of the examined antibiotic resistance genes (ARGs), were identified. The concentrations of selected gene copy numbers conferring resistance to four groups of antibiotics as well as class 1 and 2 integron-integrase genes were determined in the analyzed samples. The concentrations of six HMs and antibiotics corresponding to genes mediated resistance from 3 classes were determined. Based on network analysis, only some of the analyzed antibiotics correlated with ARGs, while HM levels were correlated with ARG concentrations, which can confirm the important role of HMs in promoting drug resistance. The samples from a wastewater treatment plant (WWTP) located an industrialized region were characterized by higher HM contamination and a higher number of significant correlations between the analyzed variables than the samples collected from a WWTP located in a less industrialized region. These results indicated that treated wastewater released into the natural environment can pose a continuous threat to human health by transferring ARGs, antibiotics and HMs to the environment. These findings shed light on the impact of industrialization on antibiotic resistance dissemination.

**Funding:** This study was supported by the Polish National Science Centre in the form of a grant awarded to GP (2017/M/NZ9/00071).

**Competing interests:** The authors have declared that no competing interests exist.

## Introduction

Industrial development is the leading cause of environmental pollution with various substances, including heavy metals (HMs), antibiotics and pesticides [1, 2]. Highly urbanized and industrialized regions are characterized by higher levels of industrial production and higher population density than regions with extensive forest cover where tourism and recreation are the predominant income-generating activities [3, 4]. Land-use type can significantly influence the composition of wastewater reaching wastewater treatment plants (WWTPs). In highly industrialized areas, wastewater is composed of municipal, industrial and hospital wastewater, whereas in less industrialized regions, industrial wastewater has a smaller share of the sewage mix that reaches WWTPs. As a result, wastewater from industrialized regions is characterized by higher concentrations of HMs than sewage generated in less industrialized areas [5, 6].

Treated wastewater released into the environment is less abundant in HMs. Heavy metals are strongly immobilized on solid particles, and they are accumulated in sewage sludge during wastewater treatment. According to the European Commission, 11.5 million tons of sludge dry solids were generated in the European Union in 2010 during wastewater treatment, and this value is expected to increase to 13 million tons in 2020 [7]. Wastewater treatment plants are not equipped with specialized systems for eliminating HMs; therefore, the removal of HMs through accumulation in sewage sludge can be regarded as a by-product of the treatment process. The use of sewage sludge in agriculture has become a cause for concern because the strong absorption of HMs to sewage sludge can pose a serious threat for the environment and public health [8]. The suitability of sewage sludge for agriculture is determined by its HM content which is analyzed and controlled based on the standards set forth by the European Council Directive 86/278/EEC [9]. The proportions of sewage sludge used in agricultural production vary across European countries. In Sweden, Poland and Germany, around 25% of the generated sewage sludge is applied in the farming sector, whereas in Spain, France and Great Britain, this figure can be as high as 70% [7].

Antibiotics are widely used in both highly industrialized and developing countries, and they are detected in wastewater [10], soil [11] and surface waters [12]. The processes of industrialization and urbanization are followed by environmental contamination with emergence pollutants as antibiotics [13]. Further development of countries is linked with the release of many contaminants to the wastewater and the aquatic environment, due to industrial activities, population growth and agricultural activities [14]. According to the European Center for Disease Prevention and Control (ECDC), Poland belongs to the group of six European countries with the highest levels of antibiotic consumption per capita [15]. Excessive and inappropriate use of antibiotics is the main cause of antibiotic resistance around the world [16]. Antibiotics reach WWTPs with human and animal feces, incorrectly disposed unused drugs, and industrial wastewater. Their concentrations can range in wastewater influents from 116 to 750 ng/L and in effluents from 52 to 85 ng/L worldwide, which increases the risk of selective resistance [17–20]. Antibiotics are degraded by hydrolysis and photodegradation, and they are adsorbed on the surface of sludge particles, which decreases antimicrobial concentrations in wastewater [21]. Numerous research studies have demonstrated that the processing technologies deployed by WWTPs do not effectively remove antibiotics from untreated wastewater [22–24].

Wastewater treatment plants are hotspots of antibiotic resistance [25], and they are regarded as direct sources of the spread of antibiotic resistance in the environment [26]. Wastewater flowing into WWTPs contains antibiotics, HMs, antibiotic-resistant bacteria (ARB) and antibiotic resistance genes (ARGs) from hospitals, households and industrial plants. Wastewater treatment systems are designed to remove organic compounds and solid

suspensions containing bacterial cells, but there are not effective technologies for eliminating genetic material, including mobile genetic elements (MGEs) [19, 27]. In addition, the WWTPs are also not capable to eliminate completely the pharmaceuticals such as antibiotics during the treatment process, literature shown the presence of these compounds in influent and effluent wastewater [28, 29].

There is a growing body of evidence to indicate that the copy numbers of ARGs can be high in wastewater with low levels of antibiotic pollution or that the concentrations of these micro-pollutants are not bound by significant correlations [30, 31]. Research has demonstrated that in addition to antibiotics, HMs [32] and pesticides [33] can also promote the exchange of genetic elements responsible for antibiotic resistance. Unlike antibiotics, HMs are not easily degraded, and they can exert long-term selective pressure when accumulated in the environment [34]. Numerous studies have revealed positive correlations between ARG and HM levels in environments exposed to anthropogenic pressure [35, 36]. High concentrations of HMs in the environment induce co-selection mechanisms which activate the expression of one or more genes encoding resistance to antibiotics and HMs under the influence of a single stressor [5]. The presence of HMs that induce selective pressure in the environment can increase resistance to antibiotics and HMs. Copper and zinc are widely used in industrial production, and they contribute to an increase in the concentrations of selected ARGs [37, 38]. According to some researchers, HMs can induce resistance to antibiotics even in environments that are free of antimicrobials [39]. Selective pressure promotes the exchange of ARGs through horizontal gene transfer (HGT) between pathogenic and non-pathogenic microorganisms or even remotely related bacteria [40]. The dissemination of ARGs via HGT is most often linked with the presence of MGEs. Mobile genetic elements such as plasmids play an important role in the transfer of ARGs through conjugation [30, 41], and according to many authors, integrons are also significantly implicated in the spread of antibiotic resistance [42]. Integrons contain a gene encoding the enzyme integrase which promotes the incorporation of genes responsible for resistance to antibiotics and HMs. These genes make up gene cassettes in the structure of integrons. Therefore, the transfer of integrons between microorganisms can lead to the acquisition of entire gene cassettes [43] and, consequently, the spread of resistance in the environment [19].

As mentioned earlier, highly industrialized regions are a significant source of pollutants such as antibiotics and HMs. Industrial activity promotes the dissemination of antibiotic resistance which poses a considerable threat for public health and the ecological balance. Limited research papers determined the influence of industrialization to the dissemination of antibiotic resistance genes and antibiotic resistance bacteria, comparing WWTPs placed in regions with different level of industrialization. Therefore, the aim of this study was to determine the correlations between the presence of pollutants such as antibiotics and HMs and the occurrence of antibiotic resistance in two WWTPs situated in regions with different levels of industrial development. The factors exerting selective pressure, which significantly contributed to the occurrence of the examined antibiotic resistance genes (ARGs), were identified.

## Materials and methods

### Research site

Samples for analysis were collected from two WWTPs located in regions with different levels of urban and industrial development and different population density. The first WWTP is situated in the Region of Warmia and Mazury (WM-WWTP), and the second WWTP—in the Region of Silesia (S-WWTP) in Poland (Europe). Silesia is the most industrialized Polish region where mining and metallurgy are the key areas of economic activity [44]. In turn,

Warmia and Mazury is referred to as the "Green Lungs of Poland", and it is characterized by low levels of urbanization, high forest cover and the absence of heavy industry [45]. The percentage of the population living in urban areas is higher in Silesia (77.6%) than in Warmia and Mazury (59.3%). The compared regions also differ in population density which is estimated at 366 persons/ $km^2$ in Silesia and only 59 persons/ $km^2$ in Warmia and Mazury [46].

Both WWTPs deploy mechanical and biological wastewater treatment systems, but an additional technology for the chemical removal of phosphorus has been implemented in S-WWTP. WM-WWTP processes wastewater from the capital city of the Region of Warmia and Mazury and four surrounding municipalities, and it has an average daily processing capacity of 35,000 $m^3$. The plant has a population equivalent (PE) of 260,361, and treated wastewater is characterized by chemical oxygen demand (COD) of 49 mg/L, and biological oxygen demand (BOD) of 6 mg/L. Industrial sewage accounts for only 8% of the total wastewater processed by WM-WWTP [47, 48]. The processing technology involves mechanical treatment systems with automated screens for separating large solids, two horizontal sand filters, and two sedimentation tanks. The biological treatment system is composed of phosphate and nitrogen removal chambers, and five multifunctional bioreactors equipped with four Passavant aerating rotors each and secondary sedimentation tanks. Sewage sludge is inactivated in closed and open digestion chambers, and it is dehydrated by a mechanical filter press. Processed wastewater is discharged to a river [49].

Wastewater from local cities and factories is supplied to S-WWTP by four trunk lines. The plant receives 32,731 $m^3$ of wastewater each day and has a PE of 403,138. Industrial sewage accounts for up to 25% of the wastewater processed by S-WWTP. Wastewater entering the plant via trunk collectors is pumped to a pre-treatment line composed of hook screens, belt conveyors and a manual screen for the separation of large solids. Pretreated wastewater is then passed through horizontal sand filters. Filtered wastewater undergoes biological treatment in the C-TECH reactor. In the first stage, phosphorus is precipitated chemically with an iron coagulant. Wastewater is then directed to the reactor where it is treated in a repeated series of aeration, sedimentation, and decantation processes. The last stage of biological treatment involves phosphorus removal, denitrification and nitrification. The generated sewage sludge is neutralized in anaerobic digestion chambers, and it is dehydrated in a high-speed decanter centrifuge and a belt filter press [50]. Schematic diagrams of the analyzed WWTPs are presented in S1 and S2 Figs in S1 File.

## Sample collection

Samples of wastewater, sewage sludge and river water were collected in June and November 2018. Wastewater and sewage sludge were sampled in different stages of treatment in both WWTPs. Samples of river water were collected upstream and downstream from the wastewater discharge point. The sampling sites are described in detail in Table 1 and S1 and S2 Figs in S1 File. Samples of wastewater and river water were collected into sterile 500 mL bottles (SIMAX, Czech Republic), and sewage sludge was sampled with the use of 100 mL plastic tubes (BioSigma, Italy). Wastewater was sampled in triplicate at hourly intervals for 24 h to obtain representative samples, and the collected hourly samples were pooled to a composite sample. Samples of river water and sewage sludge were also collected in triplicate to create complex samples. The samples were transported to the laboratory at a temperature of 4˚C.

## Determination of antibiotics

**Chemicals.** Ciprofloxacin (CIP), norfloxacin (NOR), ofloxacin (OFX), oxytetracycline (OTC), pefloxacin (PEF), sulfadimethoxine (SDM), sulfamethoxazole (SXT), sulfathiazole

**Table 1. Detailed information about samples type collected from wastewater treatment plants.**

| | WM-WWTP | | S-WWTP |
|---|---|---|---|
| Symbol | Sample | Symbol | Sample |
| WM1 | Raw wastewater | S1 | Raw wastewater |
| WM2 | Wastewater after primary clarifier | S2 | Wastewater after primary clarifier |
| WM3 | Wastewater treated in biological chamber | S3 | Wastewater after secondary clarifier |
| WM4 | Wastewater treated in multifunctional reactor | S4 | Wastewater treated in C-TECH reactor |
| WM5 | Treated wastewater | S5 | Treated wastewater |
| WM6 | River water—upstream form the effluent discharge point | S6 | River water—upstream form the effluent discharge point |
| WM7 | River water—downstream form the effluent discharge point | S7 | River water—downstream form the effluent discharge point |
| WM8 | Sewage after the open fermentation tank | S8 | Sewage sludge dewatering leachates |
| WM9 | Dewatered sewage sludge | S9 | Mechanically compacted sewage sludge |
| | | S10 | Sewage sludge after gravity thickening |

*Abbreviations as WM-WWTP stands for wastewater treatment plant situated in Warmia and Mazury Region, and S-WWTP stands for wastewater treatment plant situated in Silesia Region.

(ST) and tetracycline (TET) of high purity grade (>90%) were purchased from Sigma–Aldrich (Germany). Ciprofloxacin-D8 (CIP-D8), sulfamethoxazole-D4 (SXT-D4), ofloxacin-D8 (OFX-D8) and tetracycline-D6 (TET-D6) were obtained from Toronto Research Chemicals (Toronto, Canada). The solvents, HPLC gradient grade methanol, acetonitrile (LiChrosolv) and formic acid (98%) were supplied by Merck (Germany). Ultrapure water was obtained from a Millipore water purification system (Milli-Q water) by Merck (Germany). Ethylenediaminetetraacetic acid, $KH_2PO_4$, $MgSO_4$, NaCl, $Na_3Citrate$ and $Na_2Citrate \cdot H_2O$ was purchased from Avantor Performance Materials (Poland).

**Sample preparation.** Samples were prepared according to a previously described procedure [20]. Briefly, wastewater samples were filtered through glass fiber (GF/C, Whatman, United Kingdom) and membrane filters (0.2 μm, Sartorius, Germany). Samples of pretreated wastewater in the amount of 100 mL (diluted with water, v/v, 1:1, WM1, S1, S8) or 200 mL, containing 200 mg of ethylenediaminetetraacetic acid (EDTA) (Avantor Performance Materials, Poland) and a mixture of internal standards (0.5 mL, 1 μg/mL of CIP-D8, SXT-D4, OFX-D8 and TET-D6 in methanol) were loaded onto Oasis HLB solid phase extraction columns (3 mL, 400 mg; Waters Corp., Milford, MA). The elution was performed with 100% methanol (3×2 mL) (Avantor Performance Materials, Poland). The eluents were evaporated under a stream of nitrogen (99.999% purity, Multax, Poland) at 40°C, reconstituted in a methanol–water mixture (10:90 v/v) (1 mL) and injected (10 μL) onto an Kinetex RP-18 column (100 mm × 4.6 mm, 2.6 μm) supplied by Phenomenex (Torrance, CA, USA).

Sewage sludge samples were centrifuged at 5000 g for 5 min (MPW-352R, MPW MED. INSTRUMENTS, Poland) and the supernatant were discarded. Specimens of 5 g (WM9, S9, S10) or 10 g were placed in 50 mL polypropylene tubes (Sarstedt, Germany), combined with 50 μL of the internal standard mixture, 9 mL of 30 mM $KH_2PO_4$ and 1 mL of methanol (100%) (Avantor Performance Materials, Poland), and extracted for 20 min using 10 mL of acetonitrile (100%), with 1% formic acid and modified QUECHERS salts (4 g of $MgSO_4$, 1 g of NaCl, 1 g of $Na_3Citrate$ and 0.5 g of $Na_2Citrate \cdot H_2O$). Next, the organic layer was washed with an octadecyl sorbent (Agilent, USA), and $MgSO_4$ and centrifuged at 5000 g for 5 min. The extract was evaporated under a nitrogen stream (99.999% purity, Multax, Poland) at 40°C, reconstituted in 0.5 mL of methanol-water mixture (10:90) and injected (10 μL) onto an

Kinetex RP-18 column (100 mm × 4.6 mm, 2.6 μm) supplied by Phenomenex (Torrance, CA, USA). Data on suppliers of repeated chemicals can be found in chemical section.

## Analysis of heavy metal concentrations

The content of Zn, Pb, Ni, Cr, Co and As was determined in samples of wastewater, sewage sludge and river water. The analyses were performed by an accredited laboratory of the Institute for the Ecology of Industrial Areas in Katowice, Poland. The concentration of As in environmental samples was determined by hydride generation atomic absorption spectroscopy (HGAAS) (accreditation No. PB-05/4), and the content of the remaining HMs was determined by inductively coupled plasma atomic emission spectroscopy (ICP-OES) (PN-EN ISO 11885:2009) [51].

## Isolation of environmental DNA

Samples of wastewater and river water were passed through polycarbonate membrane filters (porosity– 0.2 μm, diameter– 47 mm) (Millipore, Merck, Germany) with the use of a vacuum pump (Millipore, Merck, Germany). The filters were placed in sterile 10 mL Falcon tubes (Eppendorf, Germany), and 5 mL of 1X PBS (Invitrogen, ThermoFisher Scientific, USA) solution was added. Samples were shaken with use of a Grant-Bio PTR-6 (Grant Instruments, UK) for 5 h at 50 rpm at room temperature (20–22˚C). The resulting suspension was transferred to sterile 2 mL microcentrifuge tubes (Eppendorf, Germany) and centrifuged (Centrifuge 5415R, Eppendorf, Germany) for 5 minutes at 4300 g at a temperature of 4˚C. The supernatant was discarded, and the pellet obtained from wastewater and river water samples was used for the isolation of genomic DNA with the DNeasy Power Water Kit (Qiagen, Germany). Genomic DNA was directly isolated from sewage sludge samples of 0.5 g each with the DNeasy Power Soil Kit (Qiagen). DNA was isolated in accordance with the manufacturer's instructions. The purity of the obtained genetic material was determined with a spectrophotometer (MultiSky, ThermoFisher Scientific, USA).

## Real-time quantitative PCR (qPCR)

The copy numbers of genes encoding resistance to beta-lactams ($bla_{TEM}$, $bla_{OXA}$, $bla_{SHV}$), tetracyclines (*tet*(A), *tet*(M)), fluoroquinolones (*qep*A, *aac*(6ʻ)-*Ib-cr*), sulfonamides (*sul*1, *sul*2), integrase genes (*int*I1, *int*I2), and the 16S rRNA gene were determined by quantitative PCR with the SYBR Green fluorescent dye. Primer sequences, qPCR conditions and amplification efficiency are presented in S1 Table in S1 File. All qPCR assays were conducted in the Roche LightCycler® 480 thermocycler (Roche Applied Science, Indianapolis, IN, USA). The reaction mix with a total volume of 15 μL was composed of 0.75 μL of the DNA matrix, 0.375 μL of each primer (10 μM), 3.75 μL of the SYBR GREEN master mix (Roche, Switzerland), and 9.75 μL of sterile water (HighPure $H_2O$, Roche, Switzerland). All samples were analyzed in triplicate. Every qPCR assay had a positive control (linearized plasmid with a specific gene insert) and a negative control (HighPure $H_2O$, Roche). Standard curves were generated by cloning the amplicon from the control to the pCR2.1-TOPO vector (Invitrogen, USA). Standard curves for qPCR were obtained in a range of $10^8$ to $10^2$ of gene copies/μL [52]. Standard curves were generated according to Applied Biosystems guidelines ("Creating Standard Curves with Genomic DNA or Plasmid DNA Templates for Use in Quantitative PCR").

## Data analysis

Data were processed statistically in the Statistica 13.3 program (StatSoft Inc.) with the use of the following non-parametric tests: Kruskal-Wallis test, Mann-Whitney U test, and Wilcoxon signed-rank test. The correlations between gene copy numbers in the analyzed samples were determined by the mean-ranks post-hoc test for multiple comparisons. The correlations between the concentrations of the examined genes and the content of HMs and antibiotics were determined by Spearman's rank-order correlation test [53]. Statistical significance was set at $p < 0.05$. A correlation matrix was developed in RStudio v. 1.2.5033 using the "corrplot" package. A heatmap was generated and Ward's cluster analysis of gene copy numbers in the evaluated sites was performed in RStudio using "heatmap.2" and "gplots" packages. A network analysis of the correlations between the analyzed variables was conducted in Gephi v. 0.9.2 [54]. A statistical analysis of antibiotic data was performed in MetaboAnalyst 4.0 [55]. The variables were log transformed to meet ANOVA assumptions. The ANOVA was followed by Tukey's post-hoc test. To better visualize antibiotic concentrations, data were normalized by dividing the results by the mean concentrations of all variables and cases (separately for S-WWTP and WM-WWTP).

# Results and discussion

## Antibiotic concentrations

The concentrations of antibiotics in influents and effluents have been widely analyzed in the literature, and antibiotic levels in different stages of wastewater treatment, including in sewage sludge, have been compared in recent studies [17, 18, 56, 57]. In the presented study, antibiotic concentrations were determined in eight (S-WWTP) and seven sites (WM-WWTP) in the analyzed WWTPs, including in sludge collectors (WM8, WM9, S9, S10). Samples of river water collected downstream and upstream from the wastewater discharge point were also analyzed. Antibiotic average concentrations in the analyzed samples are presented in Figs 1 and 2, and in S2 Table and S3-S11 Figs in S1 File. Antibiotic levels were determined in the range of $<$LOD—$1.2 \times 10^3$ ng/g. The samples from both WWTPs had the highest concentrations of fluoroquinolones, including CIP (0.0128–1179.9 ng/g), NOR ($<$LOD-319.7 ng/g), OFX (0.0047–66.1 ng/g) and PEF ($<$LOD-29 ng/g), and the lowest concentrations of sulfonamides, including SDM ($<$LOD-1.59 ng/g), SXT ($<$LOD-2.98 ng/g) and ST ($<$LOD-1.67 ng/g). Similar dependencies between antibiotic concentrations were reported by Dong et al. [58] in wastewater samples. The following abundance pattern was proposed based on antibiotic concentrations in the samples from both WWTPs:
CIP>NOR>OFX>PEF>OTC>TET>SXT>ST>SDM (S2 Table in S1 File).

In general, the concentrations of most antibiotics were significantly higher in sludge than in wastewater discharged by both WWTP and in river water (S2 Table in S1 File, Figs 1 and 2). The mean antibiotic concentrations were 3–4 orders of magnitude higher for NOR and OTC, and 2–3 orders of magnitude higher for CIP, OFX, SDM, TET and PEF. No significant differences in SXT levels were noted between sludge and the aqueous phase. Sulfathiazole was not detected in sludge, but conclusive results could not be obtained because the limit of detection was significantly higher in the analysis of sludge than the aqueous phase. The obtained results confirmed that fluoroquinolones and tetracyclines are eliminated mainly by sorption to sludge, whereas SXT are less effectively removed by this mechanism [59, 60].

Wastewater treatment significantly decreased the concentrations of CIP, OFX, NOR and ST in the aqueous phase (significant differences were observed between S1 and S5 and between WM1 and WM5) (S2 Table in S1 File, S3-S5, S10 Figs in S1 File). On average, CIP

# WM-WWTP

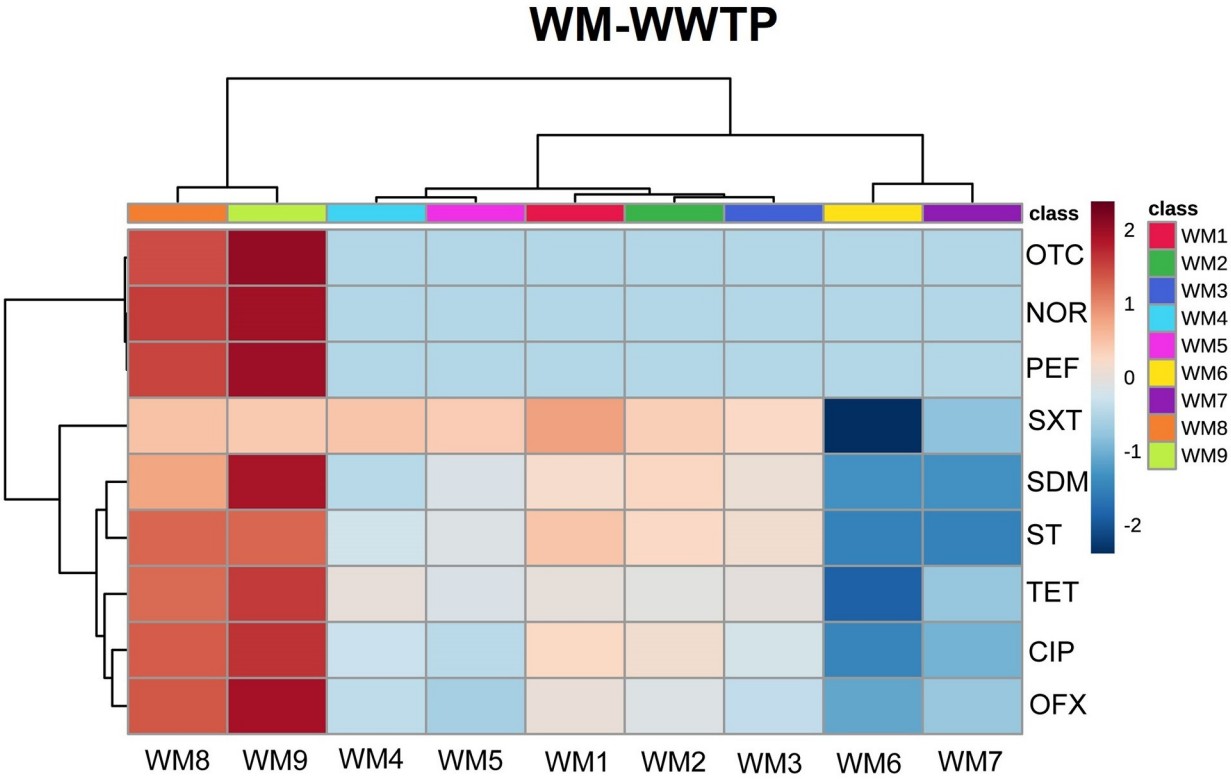

**Fig 1. Heatmap of normalized concentrations of antibiotics in different sampling points of WM-WWTP.** Red colour indicates high concentration, whereas blue colour low concentration.

concentrations were reduced 14-fold (18-fold in WM-WWTP; 10-fold in S-WWTP) (S2 Table, S3 Fig in S1 File), OFX concentrations were reduced 7-fold (6-fold in WM-WWTP; 8-fold in S-WWTP), NOR concentrations were reduced 8-fold (in S-WWTP; NOR was detected only in sludge in WM-WWTP) (S2 Table, S4 Fig in S1 File), and ST concentrations were reduced 5-fold (in both plants) (S2 Table, S10 Fig in S1 File). A significant decrease in TET concentration (100 fold) was observed only in S-WWTP (S2 Table, S11 Fig in S1 File). No changes or a minor decrease in the concentrations of SDM, OTC (7-fold decrease in S-WWTP—not significant; the results noted in WM-WWTP were below the LOD) and SXT (3-fold, significant only in WM-WWTP) were noted (S2 Table, S6, S8, S9 Figs in S1 File).

In the Mann-Whitney U test, OTC and NOR were the only antibiotics whose concentrations differed significantly ($p<0.05$) between the examined WWTPs, where S-WWTP located in industrial region was characterized with higher abundance of those pollutants (S3 Table in S1 File). Literature pointed out the concentration increase of oxytetracycline and norfloxacin in wastewater and even river sediments due to industrial activities [61–63].

Treated wastewater is discharged to rivers. Significantly higher concentrations of SXT, TET (WM-WWTP) and ST (S-WWTP) were observed in river water sampled downstream (WM7/S7) than upstream (WM6/S6) from wastewater discharge point (S2 Table, S9-S11 Figs in S1 File). The concentrations of the remaining antibiotics were low in river water. Wastewater treatment plants have been identified as hot spots of aquatic contamination with pharmaceuticals around the world [20, 64, 65].

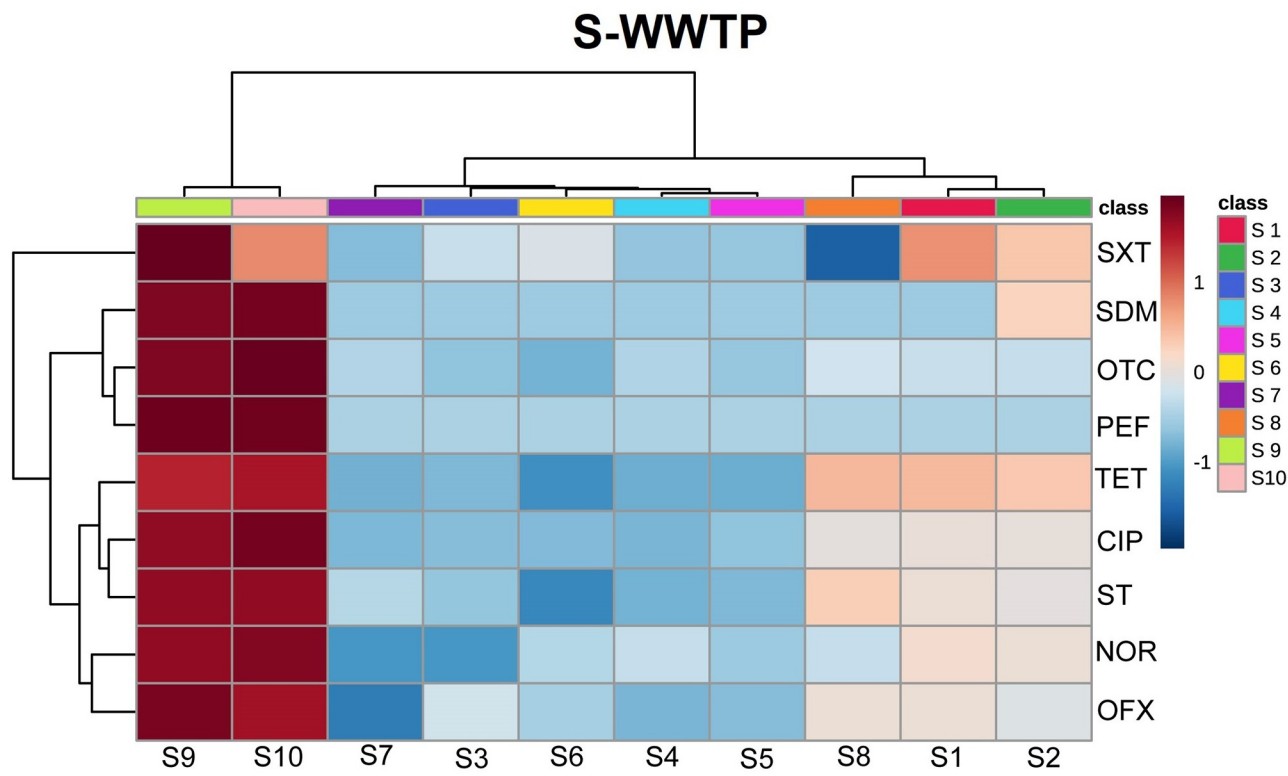

**Fig 2. Heatmap of normalized concentrations of antibiotics in different sampling points of S-WWTP.** Red colour indicates high concentration, whereas blue colour low concentration.

## Heavy metal concentrations

In this subsection, the terms of liquid samples (river water, wastewater and sewage sludge (WM8) samples) and solid samples (sewage sludge (WM9/S9/S10) samples) were used because of the method used to extract heavy metals from researched samples and the differences of concentration units. Heavy metal concentrations were determined in a range of <LOD—551 mg/L in liquid samples, and <LOD—1090.8 mg/kg DM (dry matter) in solid samples collected from both WWTPs. Heavy metal average concentrations in all samples are presented in Tables 2 and 3. Liquid and solid samples from both WWTPs were most abundant in zinc in a

**Table 2. Average concentration of analyzed heavy metals (mg/L and mg/kg d.m.) in samples from WM-WWTP.**

| | | | | WM-WWTP | | | | | | | | | | |
|---|---|---|---|---|---|---|---|---|---|---|---|---|---|---|
| | | | | June | | | | | | | November | | | |
| Sampling sites | | Zn | Pb | Ni | Cr | Co | As | Sampling sites | | Zn | Pb | Ni | Cr | Co | As |
| WM1 | mg/L | 0.247 | <0.015 | <0.009 | 0.013 | <0.006 | 0.004 | WM1 | mg/L | 0.249 | <0.015 | <0.009 | 0.008 | <0.006 | 0.002 |
| WM2 | | 0.155 | <0.015 | <0.009 | 0.010 | <0.006 | 0.003 | WM2 | | 0.138 | 0.022 | <0.009 | <0.005 | <0.006 | 0.001 |
| WM3 | | 3.018 | 0.047 | 0.039 | 0.090 | <0.006 | 0.018 | WM3 | | 2.420 | 0.032 | 0.068 | 0.104 | <0.006 | 0.005 |
| WM4 | | 3.034 | 0.045 | 0.037 | 0.084 | <0.006 | 0.015 | WM4 | | 2.640 | 0.037 | 0.068 | 0.112 | <0.006 | 0.005 |
| WM5 | | 0.065 | <0.015 | <0.009 | 0.005 | <0.006 | 0.002 | WM5 | | 0.050 | <0.015 | <0.009 | <0.005 | <0.006 | <0.0005 |
| WM6 | | 0.011 | <0.015 | <0.009 | 0.008 | 0.056 | 0.008 | WM6 | | <0.005 | <0.015 | <0.009 | <0.005 | <0.006 | <0.0005 |
| WM7 | | 0.010 | <0.015 | <0.009 | 0.008 | <0.006 | 0.011 | WM7 | | <0.005 | <0.015 | <0.009 | <0.005 | <0.006 | <0.0005 |
| WM8 | | 6.582 | 0.252 | 0.158 | 0.493 | 0.053 | 0.057 | WM8 | | 551 | 7.6 | 38.3 | 31 | 13.5 | 1.27 |
| WM9 | mg/kg d.m. | 1090.8 | 16.7 | 25.8 | 53.3 | <2 | 2.745 | WM9 | mg/kg d.m. | 934.73 | 15.72 | 8.38 | 51.58 | 8.27 | 2.69 |

**Table 3. Average concentration of analyzed heavy metals (mg/L and mg/kg d.m.) in samples from S-WWTP.**

| | | S-WWTP | | | | | | | | | | | | |
| --- | --- | --- | --- | --- | --- | --- | --- | --- | --- | --- | --- | --- | --- | --- |
| | | June | | | | | | | November | | | | | |
| Sampling sites | | Zn | Pb | Ni | Cr | Co | As | Sampling sites | | Zn | Pb | Ni | Cr | Co | As |
| S1 | mg/L | 0.299 | 0.021 | 0.017 | 0.017 | <0.006 | 0.004 | S1 | mg/L | 0.209 | 0.0156 | 0.011 | 0.006 | <0.006 | 0.001 |
| S2 | | 0.132 | <0.015 | 0.014 | 0.009 | <0.006 | 0.002 | S2 | | 0.077 | <0.015 | 0.012 | 0.010 | <0.006 | 0.004 |
| S3 | | 0.027 | <0.015 | 0.015 | 0.007 | <0.006 | <0.0005 | S3 | | 0.017 | <0.015 | <0.009 | <0.005 | <0.006 | <0.0005 |
| S4 | | 0.049 | <0.015 | 0.012 | 0.006 | <0.006 | <0.0005 | S4 | | 0.051 | <0.015 | 0.006 | <0.005 | <0.006 | <0.0005 |
| S5 | | 0.031 | <0.015 | 0.014 | 0.006 | <0.006 | 0.009 | S5 | | 0.018 | <0.015 | 0.006 | <0.005 | <0.006 | <0.0005 |
| S6 | | 0.034 | <0.015 | 0.012 | 0.008 | 0.021 | <0.0005 | S6 | | 0.011 | <0.015 | 0.003 | <0.005 | <0.006 | <0.0005 |
| S7 | | 0.022 | <0.015 | <0.009 | 0.006 | <0.006 | <0.0005 | S7 | | 0.014 | <0.015 | 0.006 | <0.005 | <0.006 | <0.0005 |
| S8 | | 3.077 | 0.088 | 0.132 | 0.119 | 0.064 | 0.026 | S8 | | 0.519 | <0.015 | 0.041 | 0.075 | 0.0160 | 0.003 |
| S9 | mg/kg d.m. | 772.3 | 23.37 | 33.74 | 39.4 | 4.6 | 1.707 | S9 | mg/kg d.m. | 706.4 | 19.0 | 25.08 | 72.96 | 4.94 | 1.15 |
| S10 | | 841.8 | 40.05 | 38.17 | 61.0 | 2.1 | 2.429 | S10 | | 767.8 | 24.96 | 24.18 | 45.09 | 0.83 | 1.70 |

concentration range of <LOD-551 mg/L and 706.4–1090.8 mg/kg DM, respectively (Tables 2 and 3). Arsenic was the least abundant HM in both liquid (<LOD-1.27 mg/L DM) and solid (1.15–2.75 mg/kg DM) samples (Tables 2 and 3). Tella et al. [66] found similar dependencies in samples of sewage sludge, whereas in the work of Olujimi et al. (2012) [67], HM concentrations were one order of magnitude higher than in the present study.

In the current research, we observed slightly higher HMs concentration in S-WWTP influent wastewater (S1:WM1) comparing to WM-WWTP (Zn 0.254:0.248; Pb 0.0183:<LOD; Ni 0.014:<LOD; Cr 0.0115:0.01 mg/L, respectively). Due to lack of similar comparisons in research papers of HMs concentration in WWTPs during treatment process in industrial and non-industrial regions, we found this issue hard to discuss.

The concentrations of Pb in sewage sludge differed between the analyzed WWTPs. The samples from WM-WWTP were less abundant in Pb, but the observed difference was not statistically significant (15.72–16.7 mg/kg d.m. in WM-WWTP and 19–40.05 mg/kg d.m. in S-WWTP) (Tables 2 and 3, S7 Table in S1 File). The above could be attributed to the fact that S-WWTP is situated in an industrial region and processes more industrial wastewater. Other authors also reported higher HMs levels in industrial wastewater and mixed municipal and industrial wastewater [6, 66].

In Poland, the limits for HMs in sewage sludge intended for agricultural use have been laid down by the Regulation of the Minister of the Environment of 2002 which transposes the provisions of the European Council Directive 86/278/EEC (European Council Directive 86/278/EEC) [9, 68]. In the current study, the concentrations of most HMs in sewage sludge did not exceed the statutory limits. The agricultural suitability of the analyzed sludge in view of their Co and As content could not be evaluated because these elements are not regulated by the above standard.

An analysis of HM concentrations in the samples collected from both WWTPs (Tables 2 and 3) revealed that treatment process affects decreasingly on HMs concentration, but without statistical significance (M-W p>0.05) in site WM1/S1 relative to site WM5/S5. Literature presented reduction of HMs concentration in industrial and non-industrial polluted wastewater, due to treatment process [69, 70]. The evacuation of treated wastewater did not increase HM concentrations in river water in site WM6/S6 relative to site WM7/S7 (M-W p>0.05). The sewage sludge was characterized by the highest HMs concentrations which differed significantly (M-W p>0.05) from those noted in wastewater samples (S5 and S6 Tables in S1 File). The Mann-Whitney U test did not reveal significant differences (p>0.05) in the HM content

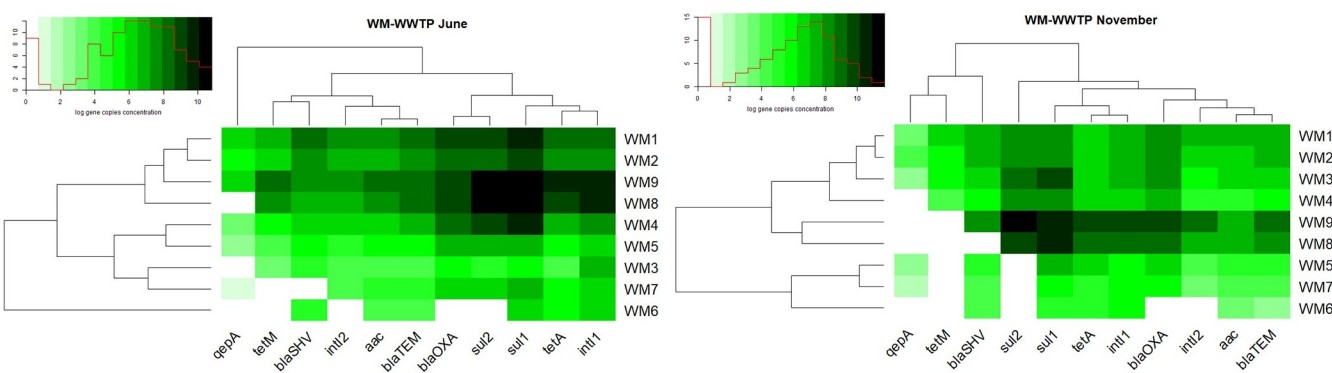

**Fig 3. Distribution of log gene copies in WM-WWTP.** Black spectrum colours indicate the highest average concentration, while white spectrum colours indicate the lowest concentration of genes copies. Dendrograms point out similarity clusters.

of samples from both WWTPs (S7 Table in S1 File). The examined WWTPs are not equipped with specialized systems for removing HMs. Heavy metals are strongly adsorbed on solid particles, and they accumulate in the sludge. According to the literature, HMs are eliminated from untreated wastewater already in settling tanks. Heavy metal removal can be regarded as a by-product of the wastewater treatment process [8, 67, 71].

## Quantification of antibiotic resistance genes

Quantitative PCR supported the identification of nine ARGs, two integrase genes and the 16S rRNA gene in the analyzed samples. The concentrations of the identified genes, excluding the 16S rRNA gene, ranged from $10^1$–$10^9$ gene copies/mL in wastewater to $10^3$–$10^{11}$ gene copies/g in sewage sludge. The average concentrations of the analyzed genes are presented in Figs 3 and 4 and in S8 and S9 Tables in in S1 File.

Genes conferring resistance to sulfonamides were most abundant in WM-WWTP. The content of *sul*1 and *sul*2 genes ranged from <LOD-2.77x$10^9$ gene copies/mL of wastewater (*sul* genes were detected in 86% of wastewater samples) to 1.22x$10^{10}$-5.23x$10^{11}$ gene copies/g of sewage sludge. In turn, *qep*A and *aac*(6')-*Ib-cr* genes encoding resistance to fluoroquinolones were least abundant at <LOD-7.7x$10^7$ gene copies/mL and <LOD-8.66x$10^7$ gene copies/g (*qep*A and *aac*(6')-*Ib-cr* were identified in 86% of wastewater samples and in 63% of

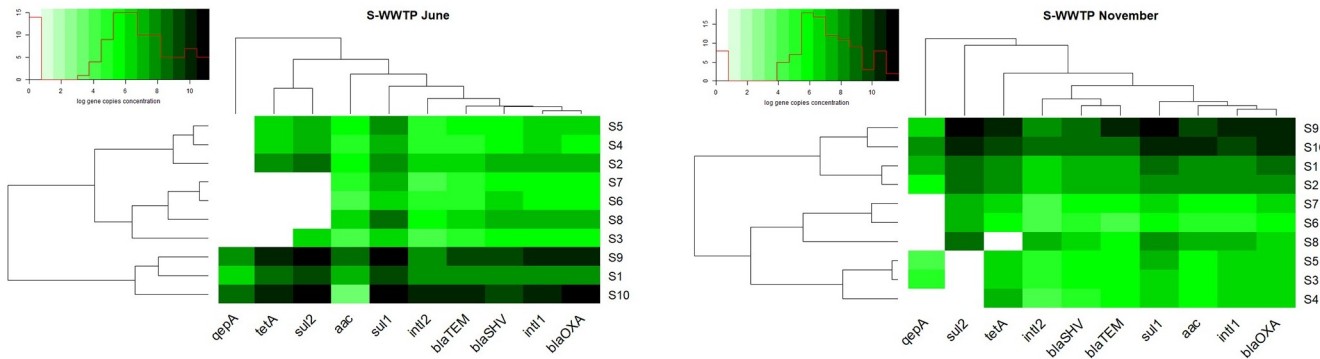

**Fig 4. Distribution of log gene copies in S-WWTP.** Black spectrum colours indicate the highest average concentration, while white spectrum colours indicate the lowest concentration of genes copies. Dendrograms point out similarity clusters.

sludge samples) (S8 Table in S1 File). In the group of fluoroquinolone-resistance genes, $qep$A was less abundant than $aac$(6')-$Ib$-$cr$ (S8 Table in S1 File). The distribution of the analyzed ARGs followed highly similar patterns in S-WWTP and WM-WWTP. Samples from S-WWTP were characterized by the highest concentrations of $sul$1 and $sul$2 genes in a range of <LOD-1.83x10$^9$ gene copies/mL ($sul$ genes were detected in 81% of wastewater samples) and 3.05x10$^{10}$-5.02x10$^{11}$ gene copies/g. The least prevalent genes were $qep$A and $aac$(6')-$Ib$-$cr$ which were determined in a concentration range of <LOD-3.58x10$^8$ gene copies/mL ($qep$A and $aac$(6')-$Ib$-$cr$ were identified in 66% of wastewater samples) and 2.7x10$^3$-1.65x10$^{10}$ gene copies/g (S9 Table in S1 File).

Chen et al. (2015) [72] examined the prevalence of 22 ARGs in wastewater and surface water samples and found that sulfonamide resistance genes were the most abundant and the only group of genes that have been identified in all samples. Le et al. (2016) [73] reported a higher concentration of the $sul$1 gene (10$^8$ copies/mL) than the $sul$2 gene (10$^7$ copies/mL) in wastewater samples, which corroborates the results of the present study. The high abundance of $sul$1 and $sul$2 genes could be attributed to their location in MGEs such as class 1 and 2 integrons [74]. Genes conferring resistance to fluoroquinolones were less abundant, which could be explained by the fact that these antibiotics are less frequently used in hospitals and the community [75]. The Mann-Whitney U test did not reveal significant differences (p>0.05) in gene concentrations between the examined WWTPs (S14 Table in S1 File).

In addition to the high content of sulfonamide resistance genes, the samples collected from both WWTPs were also highly abundant in $int$I1 and $int$I2 genes whose concentrations were determined at 7.4x10$^3$-1.7x10$^8$ gene copies/mL of wastewater and 1.24x10$^7$-3.81x10$^{10}$ gene copies/g of sewage sludge (S8 and S9 Tables in S1 File). The $int$I1 gene is frequently detected in hospital wastewater [76], whereas $int$I2 is characteristic of bacteria isolated from human and animal feces that are a potential source of this gene in wastewater [77, 78]. According to the literature, $sul$ and $int$I are closely linked, and the $sul$1 gene is frequently identified in class 1 integrons [79, 80]. Du et al. [81] pointed out that $sul$1 and $int$I1 were the most abundant genes in samples of wastewater which contains 56% of industrial wastewater.

In the group of tetracycline resistance genes, $tet$(A) was more abundant that $tet$(M) in the samples collected from WM-WWTP, whereas $tet$(M) was not identified in the samples from S-WWTP (S8 and S9 Tables in S1 File). A similar relationship was described by Chen and Zhang (2013) [82] in whose study, the concentration of $tet$(M) in municipal wastewater was three orders of magnitude lower than the content of $tet$(A). Zhang and Zhang (2011) [83] examined the prevalence of 14 genes encoding resistance to tetracyclines in samples collected from 15 WWTPs in China, and they found that $tet$(A) was the most abundant $tet$ gene.

Wastewater (WM1/S1) is treated during a series of technological processes, and treated wastewater is evacuated to rivers (WM5/S5). An analysis of the samples collected from both WWTPs revealed that the concentrations of the examined genes decreased by 2 to 3 orders of magnitude between sites WM1/S1 and WM5/S5 (S8 and S9 Tables in S1 File). According to many authors, the concentrations of resistance genes are reduced by 3 orders of magnitude on average in WWTPs that deploy the activated sludge process [84, 85]. The Kruskal-Wallis non-parametric test demonstrated a significant reduction in gene levels between samples of untreated and treated wastewater (K-W p = 0.00) (S12 and S13 Tables in S1 File). The percentage reduction of ARGs, integrase genes and the 16S rRNA gene ranged from 95.5% to 99.6% in both WWTPs (S8 and S9 Tables in S1 File).

The evaluated WWTPs evacuate treated wastewater to rivers. Effective WWTPs should not contribute to the contamination of rivers that act as receptacles of treated wastewater. An analysis of the concentrations of ARGs, integrase genes and the 16S rRNA gene in river water sampled upstream (WM6/S6) and downstream (WM7/S7) from the wastewater discharge point

revealed that the content of the studied genes increased by one order of magnitude after treatment (S8 and S9 Tables in S1 File). Despite the higher abundance of genes in river water sampled downstream from the wastewater discharge point, the observed differences were not significant (K-W p = 1.00) (S12 and S13 Tables in S1 File). An increase in pollution levels in rivers receiving treated wastewater was also frequently reported in the literature [65, 80]. The steady release of ARGs from WWTPs to rivers constitutes an important external source of antibiotic resistance in the natural environment [10, 86].

The analyzed genes were most abundant in sewage sludge samples collected from both WWTPs, and their mean concentrations were determined at $4.56 \times 10^{11}$-$1.92 \times 10^{12}$ gene copies/g in WM-WWTP and $9.65 \times 10^{11}$-$7.59 \times 10^{12}$ gene copies/g in S-WWTP (S8 and S9 Tables in S1 File). The copy numbers of ARGs and integrase genes were 2 to 3 orders of magnitude higher in sewage sludge than in all wastewater samples (S8 and S9 Tables in S1 File). Equally high concentrations of ARGs and integrase genes were noted in sewage sludge by Birošová et al. (2014) [87] and Luo et al. (2010) [88]. A statistical analysis of gene copy numbers in wastewater samples (regardless of sampling site) and sludge samples revealed significant differences, excluding the samples from sites WM1, WM2 and S1 (K-W p>0.25) (S12 and S13 Tables in S1 File). The volume of sewage sludge generated in municipal WWTPs in Poland increased by 38% between 2000 and 2018. In 2018, municipal WWTPs produced 583,000 tons of sludge dry solids, 20% of which was used in agriculture and 19.1% was thermally processed [89]. This study confirmed that sewage sludge is heavily contaminated with genetic determinants of resistance. Sewage sludge is abundant in organic matter and biogenic elements, and it is widely used as agricultural fertilizer. Research has demonstrated that long-term soil fertilization with sewage sludge increases ARG levels and promotes the spread of antibiotic resistance in soil [90].

A comparison of the relative concentrations of the studied genes and the 16S rRNA gene confirmed that *sul*1, *sul*2 and *int*I1 were most ubiquitous, whereas *qep*(A) was the least abundant gene (Figs 5 and 6). On average, gene concentrations increased by one order of magnitude between sampling sites WM1/S1 and WM5/S5 (S10 and S11 Tables in S1 File). Osińska et al. (2019) [91] also reported an increase of one order of magnitude between relative gene concentrations in untreated and treated wastewater. The results of the present study suggest that ARGs and integrase genes can be more abundant in treated wastewater than in untreated wastewater even in WWTPs equipped with effective treatment systems. These observations can be attributed to HGT which promotes the spread of ARGs to bacteria during wastewater treatment. Proia et al. (2016) [92] also observed an increase in the copy numbers of genetic elements that confer antibiotic resistance downstream from the wastewater discharge point in WWTPs with effective treatment process. Antibiotic resistance genes are evacuated with incompletely treated municipal and hospital wastewater, and they pose an environmental threat and contribute to the emergence of multidrug-resistant bacteria [93]. Despite the implementation of new wastewater processing technologies, there is a general scarcity of methods that effectively remove ARGs from wastewater, which poses a significant risk for humans and ecosystems [25]. In the current study, the average concentrations of the analyzed genes increased in samples of river water collected downstream from the wastewater discharge point in both WWTPs, but the observed increase was not statistically significant (Figs 5 and 6) (S12 and S13 Tables in S1 File).

### Impact of industrialization on antibiotic resistance—Correlation analysis

High concentrations of antibiotics and HMs in municipal and industrial wastewater induce the spread of antibiotic resistance around the world, which is why the correlations between these variables should be analyzed [10, 30].

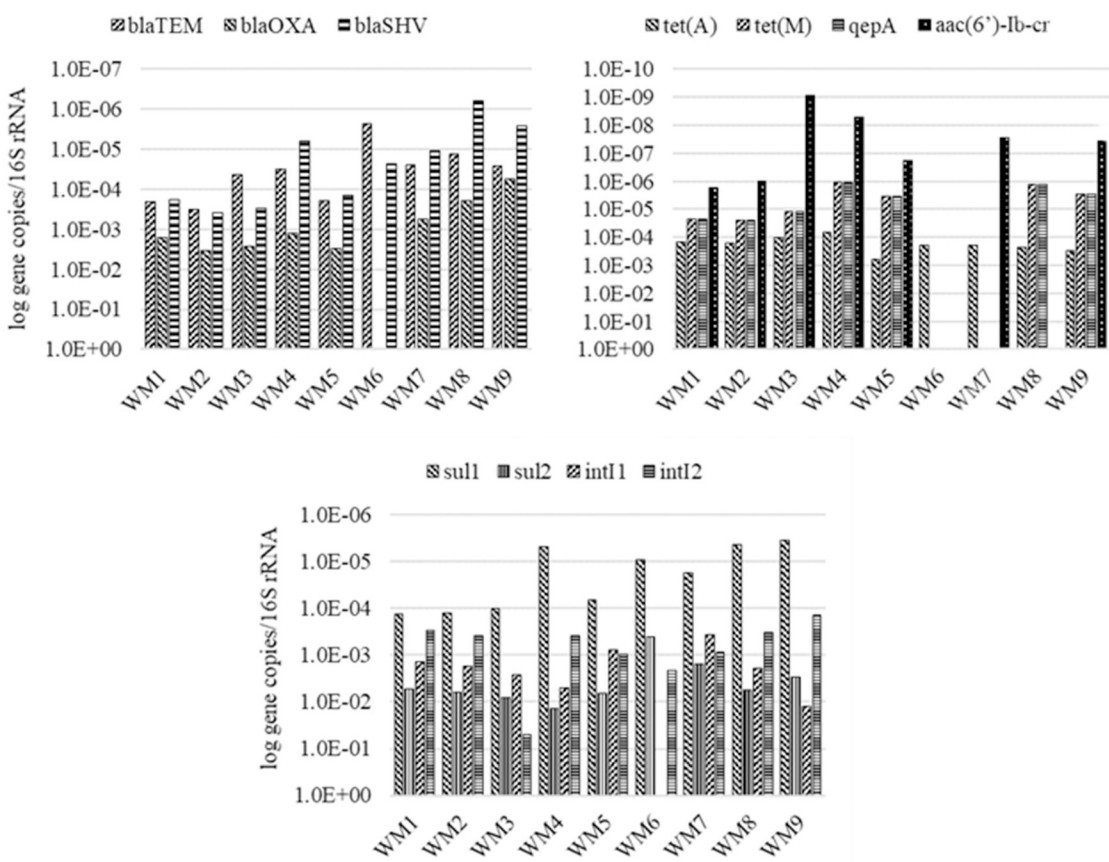

**Fig 5. Relative concentration (log gene copies/16S rRNA) of analyzed genes in WM-WWTP, both seasons.**

The correlations between the studied genes, antibiotics and HMs in the samples collected from both WWTPs are presented in Figs 7 and 8 and S12 and S13 Figs in S1 File. The results of the statistical analysis confirm the presence of strong correlations between the copy numbers of integrase genes and sulfonamide resistance genes (*sul*1 and *sul*2) in samples from both WWTPs. Di Cesare et al. (2016b) also reported strong correlations between integrase genes in wastewater. However, in the present experiment, no significant correlations were noted between sulfonamide concentrations and the abundance of sulfonamide resistance genes in any of the WWTPs. An absence of significant correlations between *sul*1 and *sul*2 genes and sulfonamide levels was also reported by Xu et al. (2015). Giebułtowicz et al. (2020) analyzed the relationship between antibiotic concentrations in wastewater and antibiotic resistance found that sulfonamides were low or average risk factors for the spread of antibiotic resistance. In the present study, no significant correlations were observed between sulfonamide levels and the copy numbers of integrase genes (*int*I1 and *int*I2). In the samples collected from both WWTPs, the abundance of *tet*(A) and *tet*(M) genes were not significantly correlated with tetracycline concentrations. An absence of significant correlations between the copy numbers of *tet* genes vs. TC and OTC concentrations was reported in samples of wastewater and river water in China [10, 94] (S12 and S13 Figs in S1 File).

The abundance of *qep*A and *aac*(6')-*Ib-cr* genes in the studied samples was correlated with fluoroquinolone concentrations. In the samples from WM-WWTP, a significant correlation was noted between *qep*A copy numbers and PEF levels, whereas in the samples from

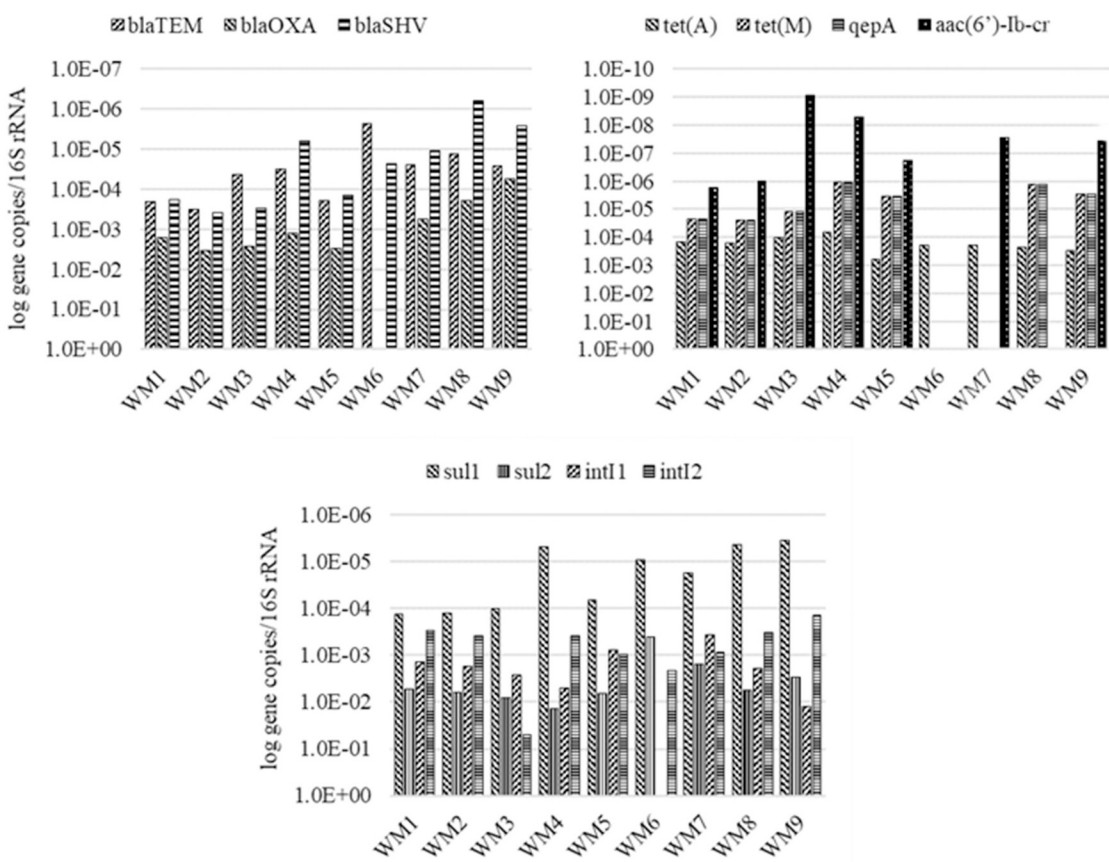

**Fig 6. Relative concentration (log gene copies/16S rRNA) of analyzed genes in S-WWTP, both seasons.**

S-WWTP, the concentrations of PEF, CIP and PEF were significantly correlated with the abundance of *qep*A and *aac*(6')-*Ib-cr* genes, respectively (S12 Fig in S1 File). Similar results were reported for fluoroquinolones by Giebułtowicz et al. (2020) who found that ciprofloxacin was a high risk factor for the dissemination of antibiotic resistance.

According to the literature [95], the prevalence of ARGs in the environment should be closely correlated with the concentrations of antibiotics that exert selective pressure on microorganisms. However, Lu et al. (2015) found that antibiotics did not significantly contribute to the dissemination of ARGs. In the present study, no significant correlations were observed between sulfonamide and tetracycline concentrations and the most prevalent genes encoding resistance to these antibiotics. The above could be attributed to the fact that tetracyclines and sulfonamides were detected at low concentrations in the analyzed samples. Despite low tetracycline and sulfonamide concentrations, the abundance of the associated ARGs was high, which could point to the presence of other selective factors in the environment, including HMs. Selective pressure could be the most problematic factor in the process of assessing the evolution of resistance. Numerous environmental factors drive variation among microorganisms, including antibiotics whose concentrations significantly change over time due to degradation. As a result, the magnitude of selective pressure is difficult to determine outside the laboratory [96]. Antibiotics are degraded much more rapidly than HMs which are retained in wastewater and promote the spread of antibiotic resistance. This observation leads to doubt as

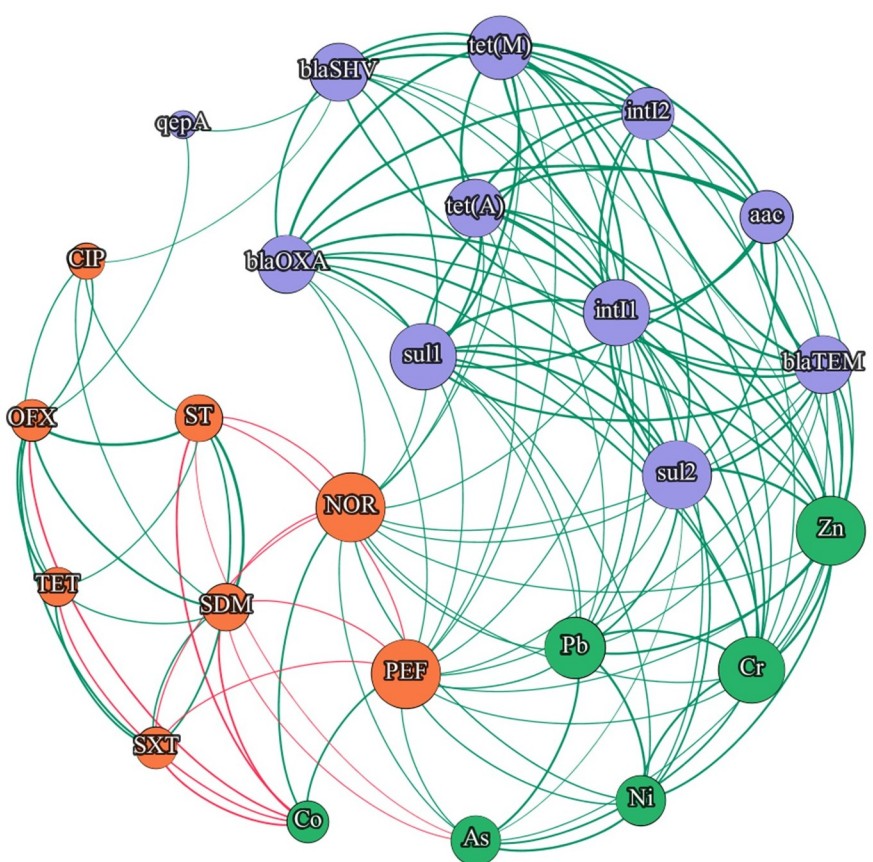

**Fig 7. Network analysis of correlated variables in WM-WWTP.** All correlations in diagram are statistically significant. Edges: green—positive correlations, red—negative correlations; Nodes: green—heavy metals, purple—genes, orange—antibiotics.

to whether high concentrations of ARGs and integrase genes should be correlated with the concentrations of antibiotics [5] or rather HMs.

Correlations between gene abundance and HM levels were more frequently noted in the samples collected from S-WWTP than WM-WWTP (Figs 7 and 8). The above could be attributed to higher mean HM concentrations in the samples from S-WWTP which is situated in a highly industrialized region. In both WWTPs, Zn, Pb, Ni and Cr were the most abundant HMs, which increased the frequency of the correlations between HMs and the analyzed genes. Chihomvu et al. [97] imply that gene encoding antibiotic resistance are located in the same position of chromosome or mobile genetic elements with heavy metal resistance genes, which highlights the position of HMs as co-selective factor for antibiotic resistance dissemination. Wastewater with industrial effluents can contain greater concentrations of emergence contaminants (as heavy metals or pharmaceuticals) by up to several orders of magnitude in comparison to municipal wastewater [63, 98, 99]. Strong correlations were noted between Cr, Ni and Zn levels and the copy numbers of *sul*1, *sul*2, *int*I1 and *int*I2 genes (r > 0.7 in all cases, p < 0.05) (S12 and S13 Figs in S1 File). Equally high correlations between ARGs and HMs were reported by Zhang et al. (2016) [100] in a study of samples from a municipal WWTP. According to the literature, the prevalence of *sul* genes is significantly correlated with HM contamination [101, 102].

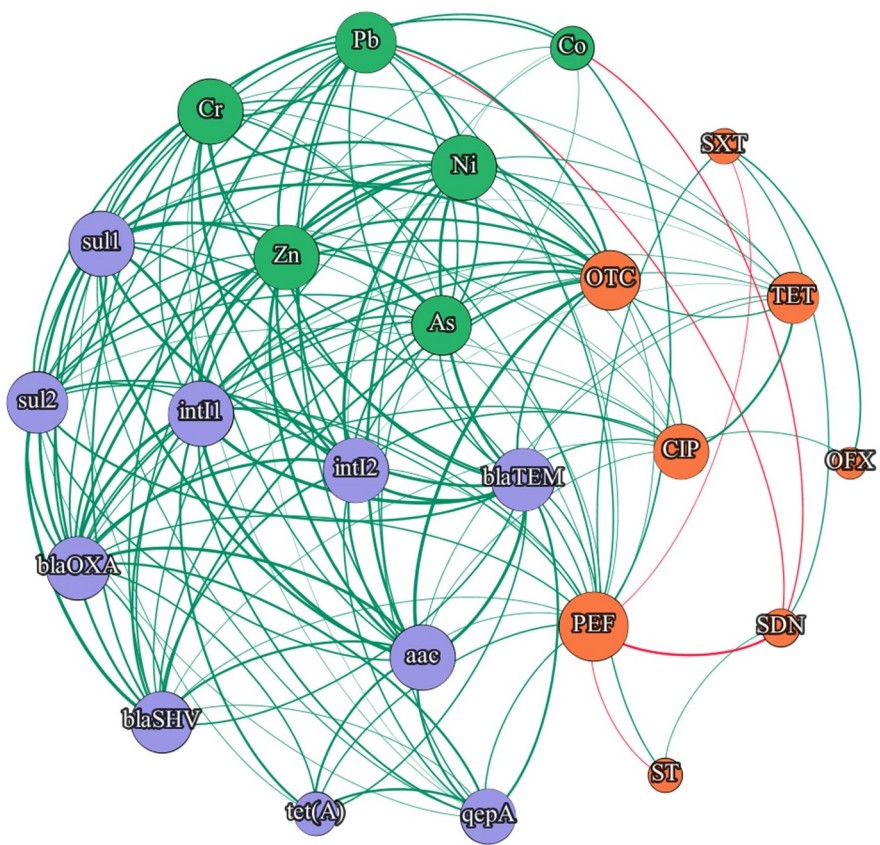

**Fig 8. Network analysis of correlated variables in S-WWTP.** All correlations in diagram are statistically significant. Edges: green—positive correlations, red—negative correlations; Nodes: green—heavy metals, purple—genes, orange—antibiotics.

In the current study, the network analysis revealed high correlations between the examined genes and HMs in the samples from WWTP located in industrial region, whereas far fewer correlations were noted between gene abundance and antibiotic concentrations (Figs 7 and 8). Regions characterized with higher levels of industrialization, release to environment high concentration of pollutants (HMs in particular), what enhance the antibiotic resistance issue through selective pressure phenomenon [33, 103–107]. Heavy metals may exert a permanent influence on the spread of ARGs and integrase genes in the environment, which can pose a serious threat for ecosystems and public health [100]. Based on the correlation analysis we can assume that antibiotics are not always the only factor responsible for the presence of ARGs in the environment. Sulfonamides have a long half-life, and their widespread use in human and veterinary medicine affects the concentration and accumulation of these antibiotics in the environment. The continuous presence of subclinical concentrations of sulfonamides in the environment can decrease the significance of these drugs as selective factors that induce antibiotic resistance [108]. As a result, antibiotic resistance can be promoted mainly by HMs. Literature presented evidence that, plasmid mediated antibiotic resistance can be induced by the selective pressure of heavy metals in the environment. Thus, beside overuse of antibiotics, environmental heavy metal pollution can lead to serious problems [107].

## Conclusions

This study investigated the correlations between the prevalence of ARGs, integrase genes and the factors that exert selective pressure and promote the spread of antibiotic resistance in wastewater, such as HMs and antibiotics. The present findings indicated that even WWTPs with effective treatment systems act as hotspots of antibiotic resistance in the environment. Sewage sludge was characterized by the highest concentrations of the analyzed chemical compounds and biological pollutants. The presence of correlations between integrase genes (*int*I1 and *int*I2), ARGs and HMs indicated that integrons (as MGEs) play a very important role in the spread of antibiotic resistance. Samples from S-WWTP which is located in a highly industrialized region were characterized by higher number of significant correlations between the investigated micropollutants and a higher concentrations of HMs. The results of the correlation analysis may assume that HMs induce antibiotic resistance in the environment. Correlation networks from both regions, shown a straight forward display that industrialization factor is a driving force for dissemination and maybe even diversification of antibiotic resistance genes in environment.

## Supporting information

**S1 File.**
(DOCX)

## Acknowledgments

Jakub Hubeny has received a scholarship from the Interdisciplinary Doctoral Program in Bioeconomy (POWR.03.02.00-00-I034/16 00) funded by the European Social Fund.

## Author Contributions

**Conceptualization:** Monika Harnisz, Ewa Korzeniewska, Grzegorz Nałęcz-Jawecki, Grażyna Płaza.

**Data curation:** Jakub Hubeny.

**Formal analysis:** Jakub Hubeny, Joanna Giebułtowicz.

**Funding acquisition:** Monika Harnisz, Grażyna Płaza.

**Methodology:** Monika Harnisz, Ewa Korzeniewska, Grzegorz Nałęcz-Jawecki, Grażyna Płaza.

**Software:** Jakub Hubeny.

**Supervision:** Monika Harnisz.

**Validation:** Jakub Hubeny, Martyna Buta, Wiktor Zieliński, Damian Rolbiecki.

**Visualization:** Jakub Hubeny, Martyna Buta, Wiktor Zieliński, Damian Rolbiecki, Joanna Giebułtowicz.

**Writing – original draft:** Jakub Hubeny, Martyna Buta, Wiktor Zieliński, Damian Rolbiecki, Joanna Giebułtowicz.

**Writing – review & editing:** Monika Harnisz, Ewa Korzeniewska, Grzegorz Nałęcz-Jawecki, Grażyna Płaza.

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
