## [Decision Letter · Decision Letter 0]

25 Mar 2021

PONE-D-21-02977

Industrialization as a Factor Enhancing Antibiotic Resistance in Wastewater, Sewage Sludge and River Water

PLOS ONE

Dear Dr. Harnisz,

Thank you for submitting your manuscript to PLOS ONE. After careful consideration, we feel that it has merit but does not fully meet PLOS ONE’s publication criteria as it currently stands. Therefore, we invite you to submit a revised version of the manuscript that addresses the points raised during the review process.

There are a number of areas for improvement identified by the reviewer. In addition, it is very important to differentiate correlation and causation and some of the observed correlation should not be explained as the cause of the observed resistance unless the mechanisms were directly examined. The revised manuscript must be thoroughly revised and all presumptive conclusions (e.g. heavy metals exert a permanent influence or HMs promoting drug resistance) need to be based on experimental data and detailed mechanistic discussions should be included.

We look forward to receiving your revised manuscript.

Kind regards,

Zhi Zhou, Ph.D.

Academic Editor

PLOS ONE

Journal Requirements:

Please include captions for your Supporting Information files at the end of your manuscript, and update any in-text citations to match accordingly. Please see our Supporting Information guidelines for more information: http://journals.plos.org/plosone/s/supporting-information.

Reviewers' comments:

Reviewer's Responses to Questions

**Comments to the Author**

1. Is the manuscript technically sound, and do the data support the conclusions?

Reviewer #1: Yes

2. Has the statistical analysis been performed appropriately and rigorously? 

Reviewer #1: Yes

3. Have the authors made all data underlying the findings in their manuscript fully available?

Reviewer #1: Yes

4. Is the manuscript presented in an intelligible fashion and written in standard English?

Reviewer #1: Yes

5. Review Comments to the Author

Reviewer #1: Please correct some minor errors specifically in the Materials and Methods section. Include the supplier and country information for all reagents and consumables used in the experimental procedures. Include the model and manufacturer information for all equipment used. Include the %/concentrations of all reagents used. This information should be included each time. The HM data should be discussed and compared with the acceptable guidelines of the country and it needs to be indicated if it comply.

6. PLOS authors have the option to publish the peer review history of their article (what does this mean?). If published, this will include your full peer review and any attached files.

Reviewer #1: **Yes: **Marthie M Ehlers

---

## [Author Response · Author response to Decision Letter 0]

8 Apr 2021

We would like to thank the Editor and Reviewer for detailed and helpful suggestions which have been taken into consideration in the revision process. Our responses to specific comments are presented below. The introduced changes and grammatical revisions have been highlighted in yellow in the manuscript.

Response to Editor suggestions

Comment: “There are a number of areas for improvement identified by the reviewer. In addition, it is very important to differentiate correlation and causation and some of the observed correlation should not be explained as the cause of the observed resistance unless the mechanisms were directly examined. The revised manuscript must be thoroughly revised and all presumptive conclusions (e.g. heavy metals exert a permanent influence or HMs promoting drug resistance) need to be based on experimental data and detailed mechanistic discussions should be included.”

We are glad of the comment, we have revised the manuscript and have made corrections. We have changed the sentences to avoid drawing straight conclusions based on the presented correlations, in “Impact of industrialization on antibiotic resistance - correlation analysis” subsection and “Conclusions” section, lines 615, 617-619, 643 and 647.

Response to Reviewer suggestions

General revision

We provided improvements through the manuscript body as reviewer requested:

- we have included suppliers information and details about reagents and consumables,

- we have included model and manufacturer information of all equipment,

- as suggested we have added significant literature,

- as suggested we have included grammatical revision 

All introduced to the manuscript changes are highlighted in yellow.

L2: Not all words in capital letters; Title does not include heavy metals

We have changed the title to “Industrialization as a source of heavy metals and antibiotics which can enhance the antibiotic resistance in wastewater, sewage sludge and river water”.

L27: Revise the aim to prevent repetition

Thank you very much for your suggestion, however, it is difficult for us to understand it and implement significant changes. We do not find in this section the repetitions indicated in the comment.

L33: Copy numbers?

Yes. We added “copy numbers” to clarify this sentence.

L36: Please clarify - was the concentration of the different antibiotics determined as well as the presence of specific resistance genes against these antibiotics which were then correlated?

To clarify the issue we have added to the sentence “that corresponded to genes encoding resistance” (L35).

L35: Clarify? or Classes?

We have changed to “classes” (L36).

L35: Please clarify and provide an exact value. L37: Please clarify - based on what is this assumption made?

Regarding to two comments, we have created changes in two sentences, explaining on what is this assumption made (L36-38).

L86: Provide values

As suggested we have added the values based on the references (L87-89).

L97: What about antibiotics? Reduce bacterial load but not resistance genes as such.

We have added the information about removal of pharmaceuticals from wastewater during treatment process (L99-102).

L130: Revise not the correct word

We have revised the mistaken word and changed it (L134). 

L201: Include a complete descriptive heading for the table

We have changed the table title to be more descriptive and added an expansion for abbreviations (L200-202).

L354: Please include how this compare with acceptable levels indicated in guidelines

As suggested we have added the comparison to acceptable levels regulated by Minister of Environment in Poland (L385-391).

---

## [Decision Letter · Decision Letter 1]

22 Apr 2021

PONE-D-21-02977R1

Industrialization as a source of heavy metals and antibiotics which can enhance the antibiotic resistance in wastewater, sewage sludge and river water

PLOS ONE

Dear Dr. Harnisz,

Thank you for submitting your manuscript to PLOS ONE. After careful consideration, we feel that it has merit but does not fully meet PLOS ONE’s publication criteria as it currently stands. Therefore, we invite you to submit a revised version of the manuscript that addresses the points raised during the review process.

Please address the comments on grammar errors, as indicted in the reviewer's comments.

We look forward to receiving your revised manuscript.

Kind regards,

Zhi Zhou, Ph.D.

Academic Editor

PLOS ONE

Journal Requirements:

Reviewers' comments:

Reviewer's Responses to Questions

**Comments to the Author**

1. If the authors have adequately addressed your comments raised in a previous round of review and you feel that this manuscript is now acceptable for publication, you may indicate that here to bypass the “Comments to the Author” section, enter your conflict of interest statement in the “Confidential to Editor” section, and submit your "Accept" recommendation.

Reviewer #1: All comments have been addressed

2. Is the manuscript technically sound, and do the data support the conclusions?

Reviewer #1: Yes

3. Has the statistical analysis been performed appropriately and rigorously? 

Reviewer #1: Yes

4. Have the authors made all data underlying the findings in their manuscript fully available?

Reviewer #1: Yes

5. Is the manuscript presented in an intelligible fashion and written in standard English?

Reviewer #1: No

6. Review Comments to the Author

Reviewer #1: The previous corrections were addressed by there are still grammar errors that need to be corrected as indicated in the attached document.

7. PLOS authors have the option to publish the peer review history of their article (what does this mean?). If published, this will include your full peer review and any attached files.

Reviewer #1: No

---

## [Author Response · Author response to Decision Letter 1]

23 Apr 2021

We would like to thank the Editor and Reviewer for detailed and helpful suggestions which have been taken into consideration in the revision process. Our responses to specific comments are presented below. The introduced changes and grammatical revisions have been highlighted in yellow in the manuscript.

Response to Editor suggestions

Comment: “Please review your reference list to ensure that it is complete and correct. If you have cited papers that have been retracted, please include the rationale for doing so in the manuscript text, or remove these references and replace them with relevant current references. Any changes to the reference list should be mentioned in the rebuttal letter that accompanies your revised manuscript. If you need to cite a retracted article, indicate the article's retracted status in the References list and also include a citation and full reference for the retraction notice.”

We are thankful for your suggestion. After a meticulous examination of our paper, we found it complete.

Response to Reviewer suggestions

General revision

We provided improvements through the manuscript body as reviewer requested:

- we have included addition suppliers information

- as suggested we have included grammatical revision 

- as suggested we have checked in references the correctness of bacterial and family names

All introduced to the manuscript changes are highlighted in yellow.

---

## [Editor Report · Decision Letter 2]

7 May 2021

PONE-D-21-02977R2

Industrialization as a source of heavy metals and antibiotics which can enhance the antibiotic resistance in wastewater, sewage sludge and river water

PLOS ONE

Dear Dr. Harnisz,

Thank you for submitting your manuscript to PLOS ONE. After careful consideration, we feel that it has merit but does not fully meet PLOS ONE’s publication criteria as it currently stands. Therefore, we invite you to submit a revised version of the manuscript that addresses the points raised during the review process.

There are still grammar errors and clarification issues. Please see the comments in the attached file from the reviewer. Comments can be found from page 64.

We look forward to receiving your revised manuscript.

Kind regards,

Zhi Zhou, Ph.D.

Academic Editor

PLOS ONE
---

## [Author Response · Author response to Decision Letter 2]

17 May 2021

We would like to thank the Editor and Reviewer for detailed and helpful suggestions which have been taken into consideration in the revision process. Our responses to specific comments are presented below. The introduced changes and grammatical revisions have been highlighted in yellow in the manuscript.

Response to Editor suggestions

Comment: “Please review your reference list to ensure that it is complete and correct. If you have cited papers that have been retracted, please include the rationale for doing so in the manuscript text, or remove these references and replace them with relevant current references. Any changes to the reference list should be mentioned in the rebuttal letter that accompanies your revised manuscript. If you need to cite a retracted article, indicate the article's retracted status in the References list and also include a citation and full reference for the retraction notice.”

We are thankful for your suggestion. After a meticulous examination of our paper, we found it complete.

Response to Reviewer suggestions

General revision

We provided improvements through the manuscript body as reviewer requested:

- we have included addition suppliers information

- as suggested we have included grammatical revision 

- as suggested we have checked in references the correctness of bacterial and family names

All introduced to the manuscript changes are highlighted in yellow.

---

## [Editor Report · Decision Letter 3]

20 May 2021

Industrialization as a source of heavy metals and antibiotics which can enhance the antibiotic resistance in wastewater, sewage sludge and river water

PONE-D-21-02977R3

Dear Dr. Harnisz,

We’re pleased to inform you that your manuscript has been judged scientifically suitable for publication and will be formally accepted for publication once it meets all outstanding technical requirements.

Kind regards,

Zhi Zhou, Ph.D.

Academic Editor

PLOS ONE
---

## [Editor Report · Acceptance letter]

26 May 2021

PONE-D-21-02977R3 

Industrialization as a source of heavy metals and antibiotics which can enhance the antibiotic resistance in wastewater, sewage sludge and river water 

Dear Dr. Harnisz:

I'm pleased to inform you that your manuscript has been deemed suitable for publication in PLOS ONE. Congratulations! Your manuscript is now with our production department. 

Kind regards, 

on behalf of

Dr. Zhi Zhou 

Academic Editor

PLOS ONE